# The genetic paradigms of dietary restriction fail to extend life span in *cep-1(gk138)* mutant of *C. elegans* p53 due to possible background mutations

Anita Goyala[1], Aiswarya Baruah[2], Arnab Mukhopadhyay[1]*

1 Molecular Aging Laboratory, National Institute of Immunology, Aruna Asaf Ali Marg, New Delhi, India,
2 Department of Agricultural Biotechnology, Assam Agricultural University, Jorhat, Assam

* arnab@nii.ac.in

**Data Availability Statement:** All relevant data are within the manuscript and its Supporting information files.

## Abstract

Dietary restriction (DR) increases life span and improves health in most model systems tested, including non-human primates. In *C. elegans*, as in other models, DR leads to reprogramming of metabolism, improvements in mitochondrial health, large changes in expression of cytoprotective genes and better proteostasis. Understandably, multiple global transcriptional regulators like transcription factors FOXO/DAF-16, FOXA/PHA-4, HSF1/HSF-1 and NRF2/SKN-1 are important for DR longevity. Considering the wide-ranging effects of p53 on organismal biology, we asked whether the *C. elegans* ortholog, CEP-1 is required for DR-mediated longevity assurance. We employed the widely-used TJ1 strain of *cep-1(gk138)*. We show that *cep-1(gk138)* suppresses the life span extension of two genetic paradigms of DR, but two non-genetic modes of DR remain unaffected in this strain. We find that two aspects of DR, increased autophagy and up-regulation of the expression of cytoprotective xenobiotic detoxification program (cXDP) genes, are dampened in *cep-1(gk138)*. Importantly, we find that background mutation(s) in the strain may be the actual cause for the phenotypic differences that we observed and *cep-1* may not be directly involved in genetic DR-mediated longevity assurance in worms. Identifying these mutation(s) may reveal a novel regulator of longevity required specifically by genetic modes of DR.

## Introduction

Dietary restriction (DR) extends longevity and imparts health benefits to a wide range of metazoans. Research over the past decades, using *Caenorhabditis elegans*, has revealed the role of evolutionarily conserved transcription factors like FOXA/PHA-4 [1–4], NRF2/SKN-1 [5], HIF1/HIF-1 [6], TFEB/HLH-30 [7], HNF4/NHR-49 [2], HSF1/HSF-1 [8] and FOXO/DAF-16 [9] in the regulation of the transcriptional landscape required to ensure long life span of DR animals. The mechanisms by which DR affects life span are diverse and depends on the mode of implementation. In *C. elegans*, DR may be implemented by genetic means as well as through

**Funding:** The research was supported in part by the National Bioscience Award for Career Development (BT/HRD/NBA/38/04/2016) and SERB-STAR award (STR/2019/000064) to AM, DBT's Twinning Programme for the NE (BT/PR16823/NER/95/304/2015) to AM and AB, and core funding from the National Institute of Immunology to AM. Some strains were provided by the Caenorhabditis Genetics Center, which is funded by National Institute of Health Office of Research Infrastructure Programs (P40 OD010440).

**Competing interests:** The authors have declared that no competing interests exist.

non-genetic dietary interventions. For example, the *eat-2* mutants that have defective pharyngeal pumping is considered the classical genetic models of DR [10]. We have recently shown that knocking down a serine-threonine kinase gene *drl-1* using RNAi leads to a DR-like phenotype that extends life and health span [2]. Non-genetically, DR can be implemented by diluting bacteria in liquid media [1, 5] or on solid media plates [9, 11], by peptone restriction [12], by feeding a non-hydrolysable glucose analog [13] or even by complete depletion of bacterial feed [14]. Interestingly but quite expectedly, the transcription factor requirements also vary with the model of DR. For example, the key transcription factor required in DR regime on solid plates is DAF-16 while for *eat-2* mutants, *drl-1* KD worms and for bacterial dilution in liquid, PHA-4 is required [1, 2, 9]. On the other hand, a DR regime in liquid media on solid support requires SKN-1 [5] while increased life span on complete removal of food is dependent on HSF-1 [14]. These observations point to the complex modalities of gene regulation brought about by nutrient signalling and DR implemented by various regimes that still need to be elucidated, including identifying new transcriptional regulators.

The p53 protein is a well-known tumour suppressor that has an important role in maintaining genome integrity by inducing DNA repair, cell cycle arrest and apoptosis in response to genotoxic stress [15]. Various model organisms like mice [16, 17], flies [18, 19] and worms [20, 21] have been exploited to investigate the role of p53 in the aging process. In *C. elegans*, CEP-1 is the functional ortholog of p53. Interestingly, knocking down *cep-1* by mutation or RNAi leads to a small but significant increase in life span, in a context dependent manner [20, 21]. The *cep-1(gk138)* allele has been widely used in studies on *cep-1*. The TJ1 strain was prepared by backcrossing *cep-1(gk138)* ten times with wild-type N2 [21] and is available from the Caenorhabditis Genetics Center (CGC). In this study, we asked whether TJ1 *cep-1(gk138)* produced life span extension when grown under different DR regimes. We found that the genetic modes of DR, namely *drl-1* KD and *eat-2* mutant, failed to extend life span in TJ1. However, non-genetic DR regimes, namely BDR and 2-DOG were unaffected. We show that under genetic DR, two important cellular processes, autophagy induction and cytoprotective response, through the activation of the xenobiotic detoxification program (cXDP) gene expression, fail to get upregulated in TJ1 *cep-1(gk138)*. Also, TJ1 differentially affects the two genetic models of DR in terms of regulation of fat storage. However, we found that after backcrossing the TJ1 allele 2 times, or on using two other mutant alleles, life span extension on *drl-1* RNAi became independent of the absence of *cep-1*, suggesting that background mutation(s) in the TJ1 strain may suppress genetic DR-mediated longevity extension. Interestingly, *cep-1(gk138)* TJ1 as well as the 12X backcrossed mutant consistently enhanced dauer formation in the insulin signalling defective *daf-2(e1370)*, suggesting that it may be a *bona fide* property of *cep-1*. Since TJ1 differentially influences the two genetic models of DR, it will be interesting to identify the background mutation(s).

## Results

### *Cep-1(gk138)* (TJ1) suppresses life span of genetic paradigms of DR

The *gk138* allele (TJ1; 10x backcrossed) was obtained from the Caenorhabditis Genetics Center (CGC) and has a 1660 bp deletion in the *cep-1* gene. In our lab, we are characterizing a genetic model of DR where knocking *drl-1* kinase gene leads to a DR-like phenotype, resulting in a dramatic increase in life span [2]. We found that *drl-1* KD worms failed to show life span extension in TJ1 *cep-1(gk138)* (Fig 1A) [to be called *cep-1(gk138)*]. The *eat-2* mutants represent the classical genetic models of DR [10]. We compared the life span of *eat-2(ad1116)* with the long-lived *eat-2(ad1116);cep-1(gk138)* and found that the life span of the former is completely suppressed (Fig 1B). Similar results were observed with another allele, the *eat-2(ad465)*

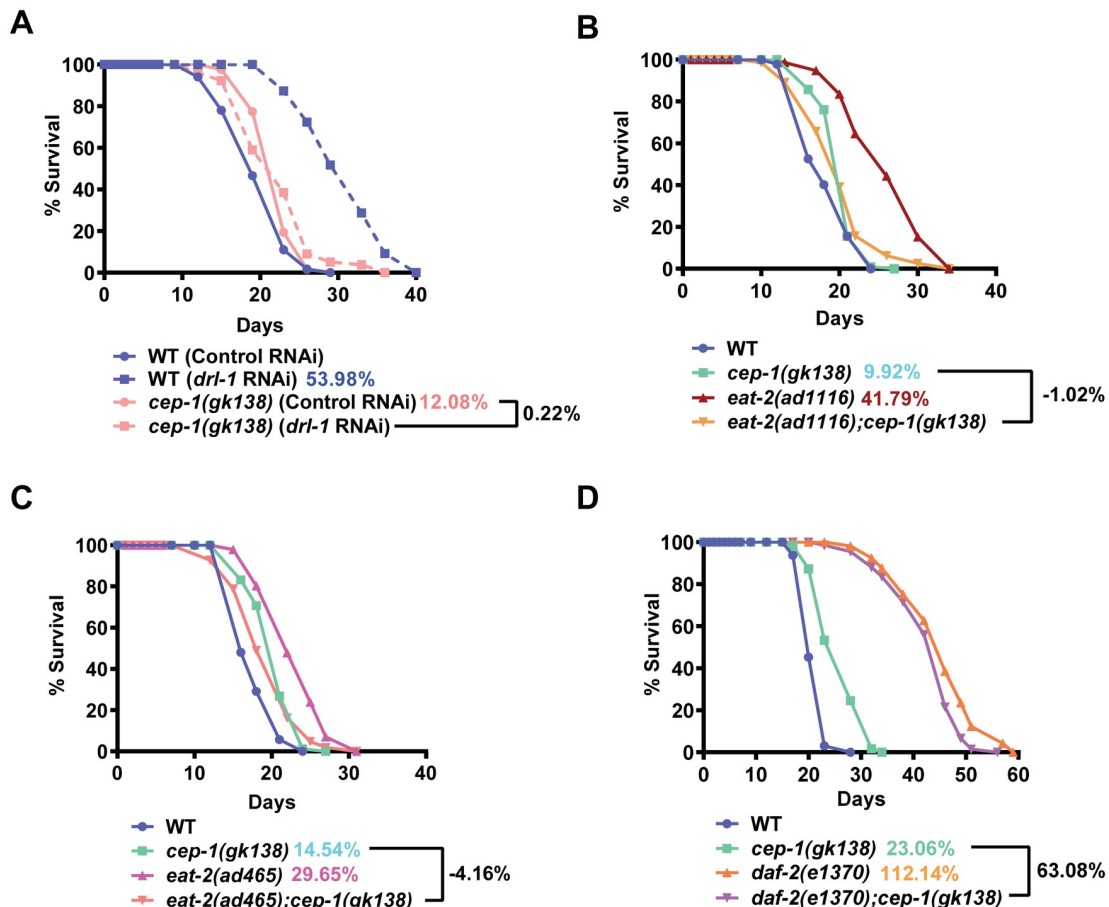

**Fig 1. In *cep-1(gk138)*, genetic paradigms of Dietary Restriction fail to extend life span.** (A) The life span extension upon *drl-1* KD in WT is suppressed when *cep-1(gk138)* is used. (B) The extended life span of *eat-2(ad1116)* is supressed in *eat-2(ad1116);cep-1(gk138)*. (C) The extended life span of *eat-2(ad465)* is supressed in *eat-2(ad465);cep-1(gk138)*. (D) The life span of *daf-2(e1370)* is partially suppressed when combined with *cep-1(gk138)*, as in *daf-2(e1370);cep-1(gk138)*. Life spans were performed at 20 ˚C. Details of life span are provided in S1 Table. Mantel-Cox log rank test using OASIS software available at http://sbi.postech.ac.kr/oasis [22] was used for statistical analysis. Life spans in 1B and 2B have identical controls as they were part of the same experiment (**also see** S1 Fig).

(Fig 1C). Together, we found that TJ1 *cep-1(gk138)* mutant suppresses life spans of two genetic paradigms of DR, the *eat-2* mutants as well as the worms grown on *drl-1* RNAi.

We next asked whether other longevity pathways are influenced by *cep-1(gk138)*. We analysed the effect on the long life span of the reduced Insulin/IGF-1 signalling (IIS) mutant, *daf-2(e1370)*. The *daf-2(e1370);cep-1(gk138)* showed a partial reduction compared to the life span of *daf-2(e1370)* (Fig 1D), showing that the effect was more robust in case of DR. Interestingly however, the dauer formation of *daf-2(e1370)* was dramatically enhanced in *daf-2(e1370);cep-1(gk138)* (S1 Fig). This shows that life span and dauer development is differentially regulated in the IIS pathway mutant in combination with TJ1 *cep-1(gk138)*.

DR may be initiated non-genetically in worms either by diluting the bacterial feed (BDR) [1] or by using a non-hydrolysable glucose analog, 2-deoxyglucose (2-DOG) [13]. We asked if the *cep-1(gk138)* disrupts the life span extension of the non-genetic paradigms of DR as well. We observed that bacterial dilution also generated the typical bell-shaped curve in the BDR assay of *cep-1(gk138)*, as seen in WT worms, when average life spans were plotted against the bacterial dilutions (Fig 2A). Likewise, on supplementation of 2-DOG, *cep-1(gk138)* showed life span

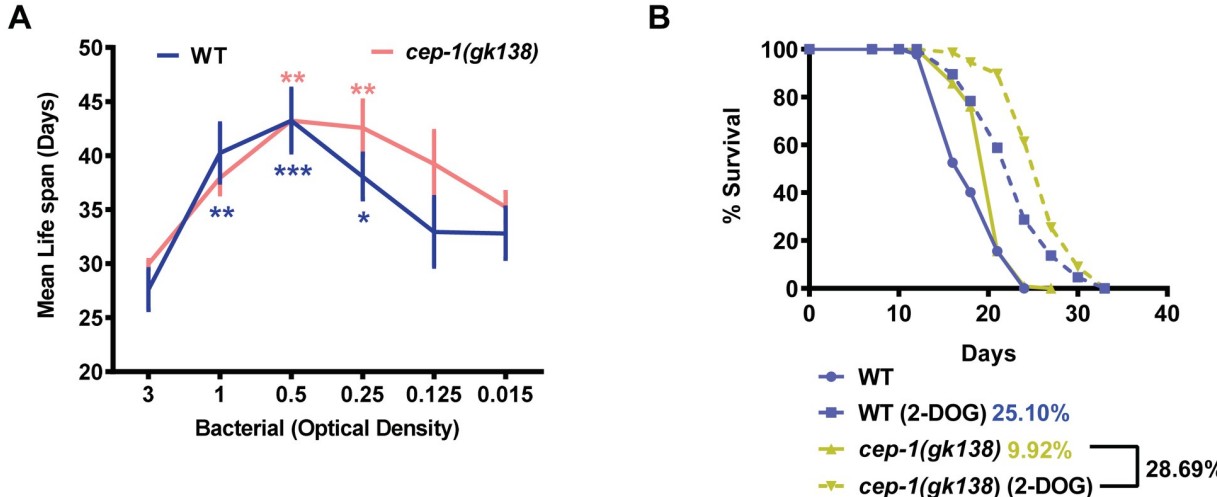

**Fig 2. The life span extension on non-genetic modes of DR remain unaffected in *cep-1(gk138)*.** **(A)** A bell-shaped curve is observed when WT or *cep-1(gk138)* worms are grown on different dilutions of bacterial feed and the average life span plotted against the $OD_{600}$. Data are presented as mean values ± SEM. N = 3 independent experiments. Two-way Annova was used for statistical analysis. *$P \leq 0.05$, **$P \leq 0.01$, ***$P \leq 0.001$. **(B)** Supplementation of 2-DOG extends life span in *cep-1(gk138)* mutant similar to WT. Life span were performed at 20˚C and details are provided in S1 Table. Mantel-Cox log rank test using OASIS software available at http://sbi.postech.ac.kr/oasis [22] was used for statistical analysis.

extension similar to WT worms (Fig 2B). Overall, both the non-genetic DR paradigms in TJ1 *cep-1(gk138)* could increase life span, indicating that it only affects the genetic modes of DR.

## Autophagy fails to be upregulated in *cep-1(gk318)* (TJ1) undergoing DR

A prominent marker for long-lived mutants is increased autophagy. DR-induced autophagy requires the transcription factors FOXA/PHA-4 and HLH-30, implying that autophagy is transcriptionally regulated during DR [7, 23]. We asked whether autophagy is mis-regulated in the TJ1 *cep-1(gk138)* undergoing the genetic paradigms of DR, providing a possible reason for life span suppression. We observed that the increase in autophagosome number, as determined by the number of puncta in the seam cells of the L3 stage *lgg-1*::*gfp* transgenic worms, after knocking down *drl-1* in WT background, was completely suppressed in *cep-1(gk138)* (Fig 3A, S2A Fig). Next, we found that the increased puncta in *eat-2(ad1116);lgg-1*::*gfp* was also suppressed in *eat-2(ad1116);cep-1(gk138);lgg-1*::*gfp* (Fig 3B, S2B Fig). Basal levels of autophagy were unchanged in *cep-1(gk138)*, compared to WT. This implies that *cep-1(gk138)* influences autophagy in the two genetic models of DR.

We next followed up this observation by determining the expression levels of four of the important autophagy genes (*lgg-1*, *bec-1*, *vps-34*, and *unc-54*) under control and *drl-1* KD conditions, using wild-type and *cep-1(gk138)*. We observed a transcriptional up-regulation of *lgg-1* and *vps-34* on knock-down of *drl-1* as compared to control RNAi, and this increase was reduced in *cep-1(gk138)* (Fig 3C). To biochemically determine the level of autophagy up-regulation, the extent of PE-LGG-1 formation was evaluated using western blot analysis. We observed that on *drl-1* KD, there was an increase in the PE-LGG-1 band intensity, corresponding to the increased autophagosome numbers seen in the puncta assay. In line with the autophagosome assay, the PE- LGG-1 band intensity reduced in *cep-1(gk138)* as compared to wild-type on *drl-1* KD (Fig 3D).

In order to find whether the regulation of autophagy in *cep-1(gk138)* was specific to the modes of DR used in this study, we determined whether autophagy induction in *daf-2(e1370)*

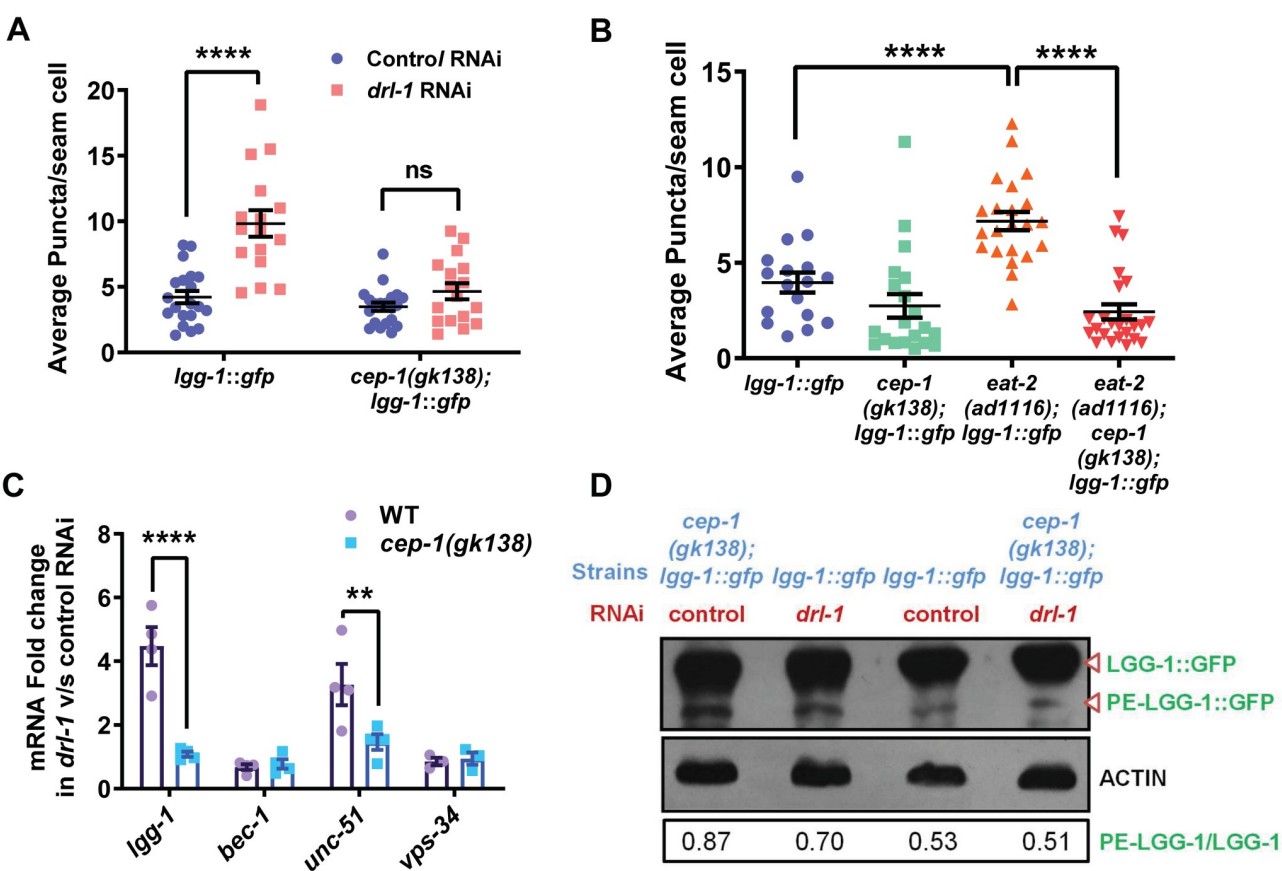

**Fig 3. *Cep-1(gk138)* regulates autophagy during DR. (A)** Increased autophagosome formation in *lgg-1*::*gfp* grown on *drl-1* RNAi is suppressed in *cep-1 (gk138); lgg-1*::*gfp*. Quantification of GFP puncta for one of three independent experiments is shown. Data are presented as mean values ± SEM. No. of animals analysed, n≥16. Two-way Annova with Sidak's multiple comparison tests was used for statistical analysis. \*\*\*\**P*≤0.0001, ns = non-significant. **(B)** The increased autophagosome formation in *eat-2(ad1116)* is suppressed in *eat-2(ad1116);cep-1(gk138)*. Quantification of GFP puncta for one of three independent experiments is shown. Data are presented as mean values ± SEM. No. of animals analysed, n≥17. Unpaired two-tailed t-test with Welch's correction was used for statistical analysis. \*\*\*\**P*≤0.0001. Source data is provided in S1 File. **(C)** Transcript levels of autophagy genes, *lgg-1*, *bec-1*, *unc-51*, *vps-34* in WT and *cep-1(gk138)* on control and *drl-1* RNAi. Data are presented as mean values ± SEM. N≥3 independent experiments. Two-way Annova with Sidak's multiple comparisons test was used for statistical analysis. \*\**P*≤0.01, \*\*\*\**P*≤0.0001. **(D)** Western blot using anti-GFP antibody to detect the unmodified LGG-1 or PE-LGG-1. The *lgg-1*::*gfp* or *cep-1(gk138);lgg-1*::*gfp* worms were grown on control or *drl-1* RNAi. Densitometric quantification of protein bands were performed using ImageJ and ratio of PE-LGG-1/LGG-1 is shown. β-ACTIN was used as a loading control. One representative blot out of the two independent experiments is shown. Experiments were performed at 20 ˚C. Uncropped blots are provided in S1 File (**also see** S2 Fig).

is also influenced by the TJ1 *cep-1(gk138)*. In agreement with previous data [24], we also found that reduced IIS led to increased autophagic puncta. However, this increase was maintained in the *cep-1(gk138)* background (S2C–S2E Fig). These observations show that TJ1 *cep-1(gk138)* regulates autophagy in response to genetic modes of DR, but not on lowering IIS signalling.

## Fat storage is differentially affected in the two genetic DR models by *cep-1 (gk138)*

One of the hallmarks of DR is metabolic reprogramming towards increased fatty acid oxidation that leads to depletion of stored fat [2]. We asked whether *cep-1(gk138)* would have differences in depletion of stored fat in the two genetic models of DR. We found that on *drl-1* KD in both WT and TJ1 *cep-1(gk138)*, fats stores were depleted, as measured by Oil Red O staining of fixed worms (Fig 4A). Surprisingly, in the *eat-2(ad1116);cep-1(gk138)*, the reduced fat stores of

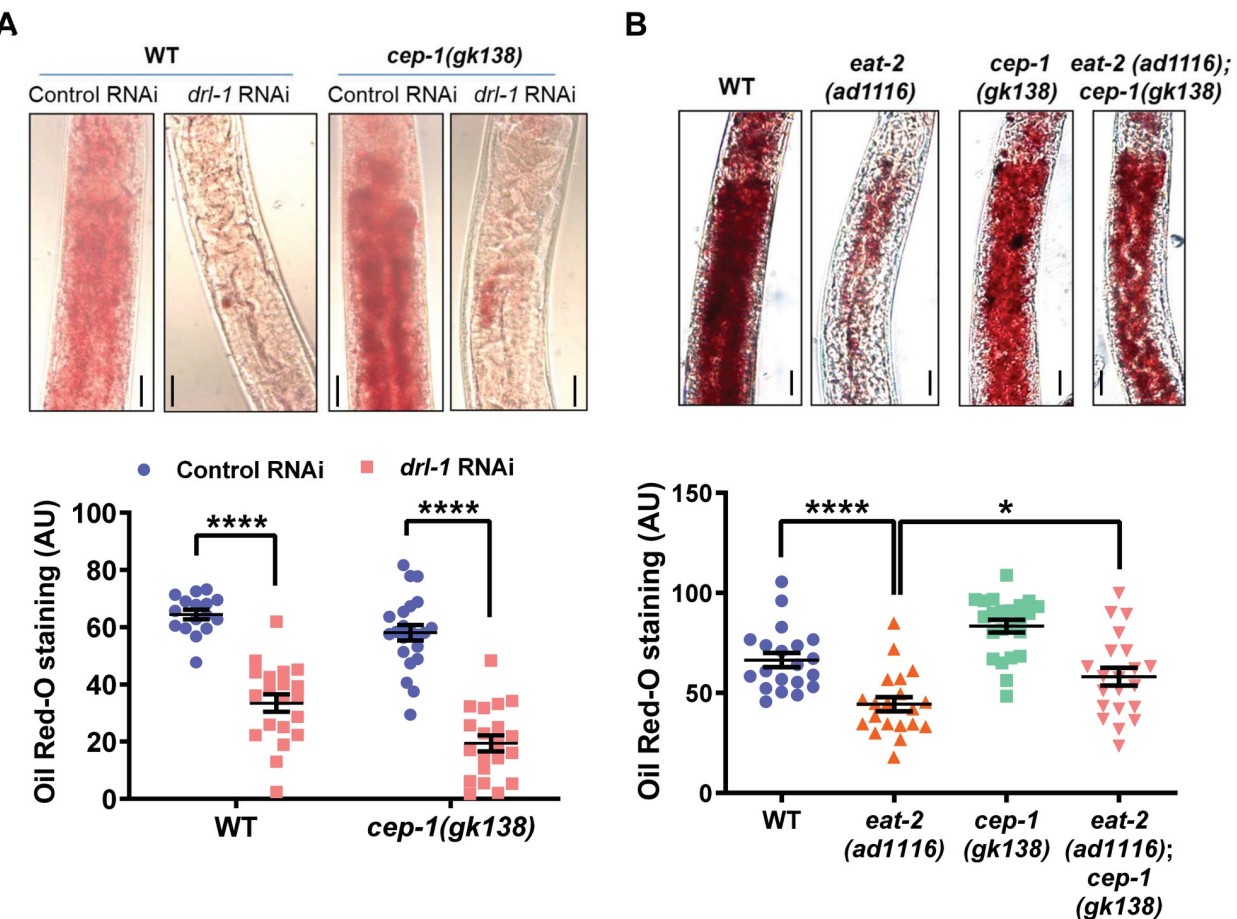

**Fig 4. Fat storage is differentially regulated by *cep-1(gk138)*.** (**A**) Oil Red O staining of WT or *cep-1(gk138)* grown on control or *drl-1* RNAi. Lowering of fat storage takes place in both the strains grown on *drl-1* RNAi. Representative images (upper panel) and quantification (lower panel) for one of three independent experiments is shown. Images were captured at 400X magnification. Scale bar is 20 μm. Data are presented as mean values ± SEM. No. of animals analysed, n≥16. Two-way Annova with Sidak's multiple comparisons test was used for statistical analysis. ****$P \leq 0.0001$. (**B**) Oil Red O staining of *eat-2(ad1116)* or *eat-2(ad1116);cep-1(gk138)*. Lowering of fat storage in *eat-2(ad1116)* is attenuated in *eat-2(ad1116);cep-1(gk138)*. Representative images (upper panel) and quantification (lower panel) for one of three independent experiments is shown. Images were captured at 400X magnification. Scale bar is 20 μm. Data are presented as mean values ± SEM. No. of animals analysed, n≥20. Unpaired two-tailed *t*-test with Welch's correction was used for statistical analysis. ****$P \leq 0.0001$, *$P \leq 0.05$. Experiments were performed at 20 ˚C. Representative images presented in Fig 4B and S3 Fig are from the same experiment and thus, have the same control panels. Source data is provided in S1 File (**also see** S3 Fig).

*eat-2(ad1116)* was partially restored (Fig 4B). Similar observations were made for *eat-2 (ad465);cep-1(gk138)* (S3 Fig). Together, TJ1 *cep-1(gk138)* responds differently to the two genetic modes of DR.

## Suppression of cytoprotective gene activation in *cep-1(gk138)* undergoing genetic DR

Previous study from our lab has shown that *drl-1* KD leads to upregulation of the cytoprotective xenobiotic detoxification pathway (cXDP) genes to ensure DR-mediated life span enhancement [2]. This up-regulation requires the conserved transcription factors like PHA-4, NHR-8 and AHR-1. Since CEP-1 is also known to regulate detoxification genes like *gst-4* [25], we asked whether the cXDP genes are optimally upregulated in TJ1 *cep-1(gk138)* undergoing

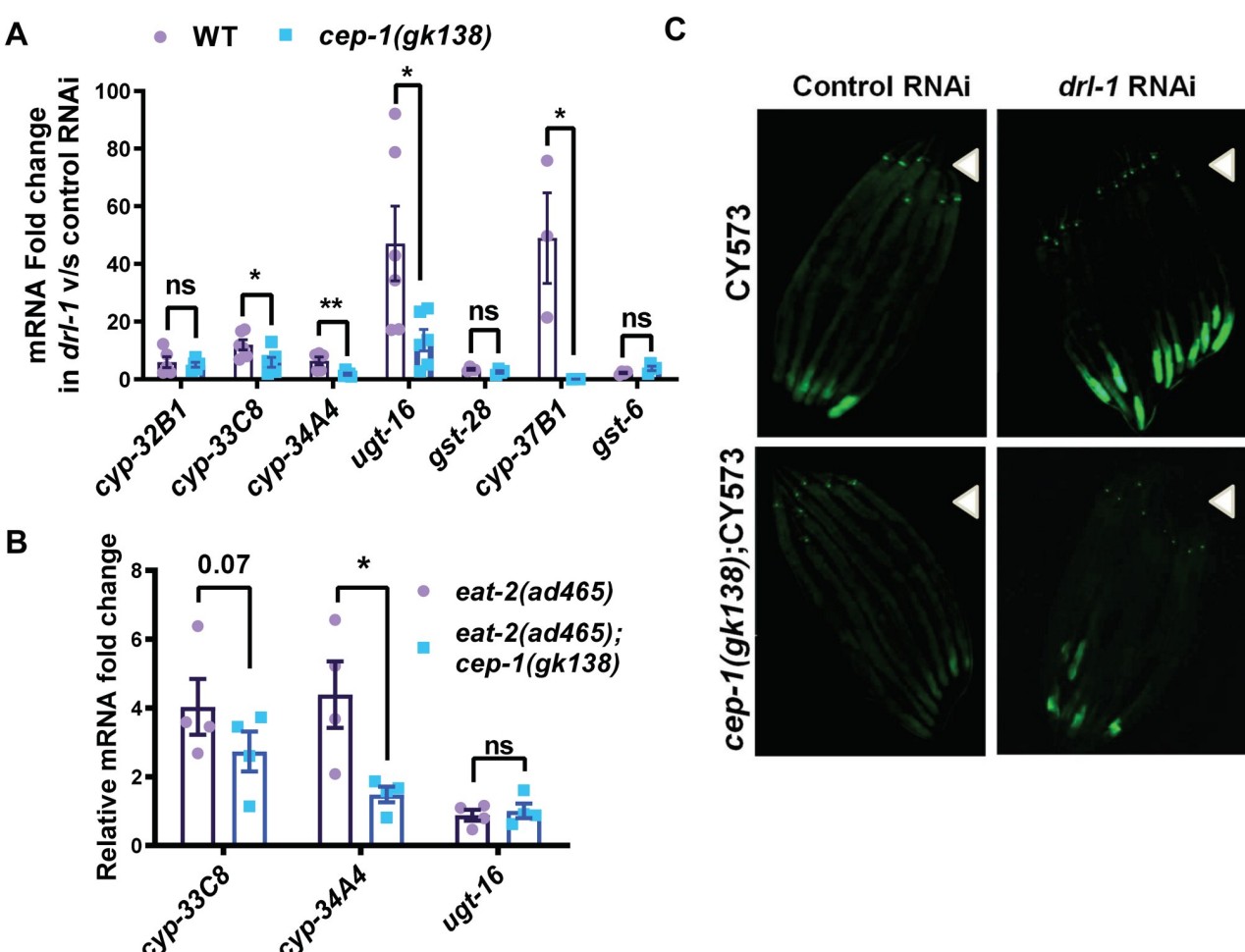

**Fig 5. The increased expression of cXDP genes in *cep-1(gk138)*, undergoing genetic DR, is partially attenuated. (A)** The mRNA levels of cXDP genes are increased in WT grown on *drl-1* RNAi, but partially suppressed in *cep-1(gk138)*. Data are presented as mean values ± SEM. N≥3 independent experiments. Unpaired two-tailed *t*-test was used for statistical analysis. *$P$≤0.05, **$P$≤0.01, ns = not significant. **(B)** The increased levels of cXDP genes in *eat-2(ad465)* is reduced in *eat-2(ad465);cep-1(gk138)*. Data are presented as mean values ± SEM. N≥3 independent experiments. Unpaired two-tailed *t*-test was used for statistical anaysis. *$P$≤0.05, ns = not significant. **(C)** Representative images of one of two independent experiments, showing increase in *pcyp-35B1:gfp* expression on *drl-1* KD in WT, which abrogated in *cep-1(gk138)*. n≥20. Multiple overlapping images were captured at 100X to cover the length of whole worm and then stitched together to generate a contiguous image. Arrow heads point towards the head of the worms. Experiments were performed at 20 ˚C.

genetic modes of DR. Quantitative RT-PCR showed that the phase I and phase II cXDP were up-regulated in wild-type on *drl-1* KD. However, the extent of upregulation of some of these genes, like *cyp-33*, *cyp-35*, *cyp-37* and *ugt-16* was significantly reduced in *cep-1(gk138)* grown on *drl-1* RNAi. Other genes, like *cyp-32* and *gst-28* were downregulated but not significantly (Fig 5A). The transcript levels of three of these genes were also determined in *eat-2* mutant and two were found to be reduced in the TJ1 background (Fig 5B).

We also used the transcriptional reporter-containing transgenic strain, *Pcyp- 35B1::gfp* that has the promoter of *Cyp-35B1*, a gene of phase I XDP, driving the expression of GFP in the intestine. When *drl-1* is knocked down in *Pcyp-35b1::gfp*, the expression of GFP increases but this was suppressed in *cep-1(gk138);Pcyp35b1::gfp* (Fig 5C). Together, these results show that the optimal expression of cXDP genes during DR is hampered in TJ1 *cep-1(gk138)*.

## Background mutation(s) in TJ1 *cep-1(gk138)* may be responsible for suppressing DR life span

The TJ1 strain obtained from CGC is 10x backcrossed. We observed that while the original 10x (Fig 1A) as well as the 11x backcrossed [the CGC TJ1 *cep-1(gk138)* backcrossed once] strains showed complete suppression of life span when grown on *drl-1* RNAi (Fig 6A), the 12X strain [the CGC TJ1 *cep-1(gk138)* backcrossed twice] failed to do so (Fig 6B). We reordered the stains from CGC along with VC172 that is 0X backcrossed. We found that the newly acquired TJ1 strain still led to complete suppression of *drl-1* RNAi life span extension (Fig 6C) while similar observation was not seen with the VC172 (Fig 6D). Further, we tested two other alleles of *cep-1*, namely *cep-1(ep347)* and *cep-1(lg12501)* as well as another 12X backcrossed line generated earlier [20] and found that life span was not suppressed as before (Fig 6E and 6F, S4A Fig). Together, we believe that a background mutation(s) in the 10X TJ1 strain of *cep-1(gk138)* suppresses the positive effects of the two different genetic modes of DR. Interestingly though, the dauer enhancement phenotype of *daf-2(e1370)* is still retained in the *daf-2(e1370); cep-1(gk138)*(12X) (S4B Fig), suggesting either that this phenotype is specific to *cep-1* or that the mutation is closely linked to *daf-2* locus. Future studies to characterize and identifying the mutation(s) may lead to finding novel regulators of DR life span.

## Discussion

P53 is a multi-functional transcription factor that directs the regulation of various biological processes such as DNA repair, apoptosis, cellular senescence and autophagy in response to

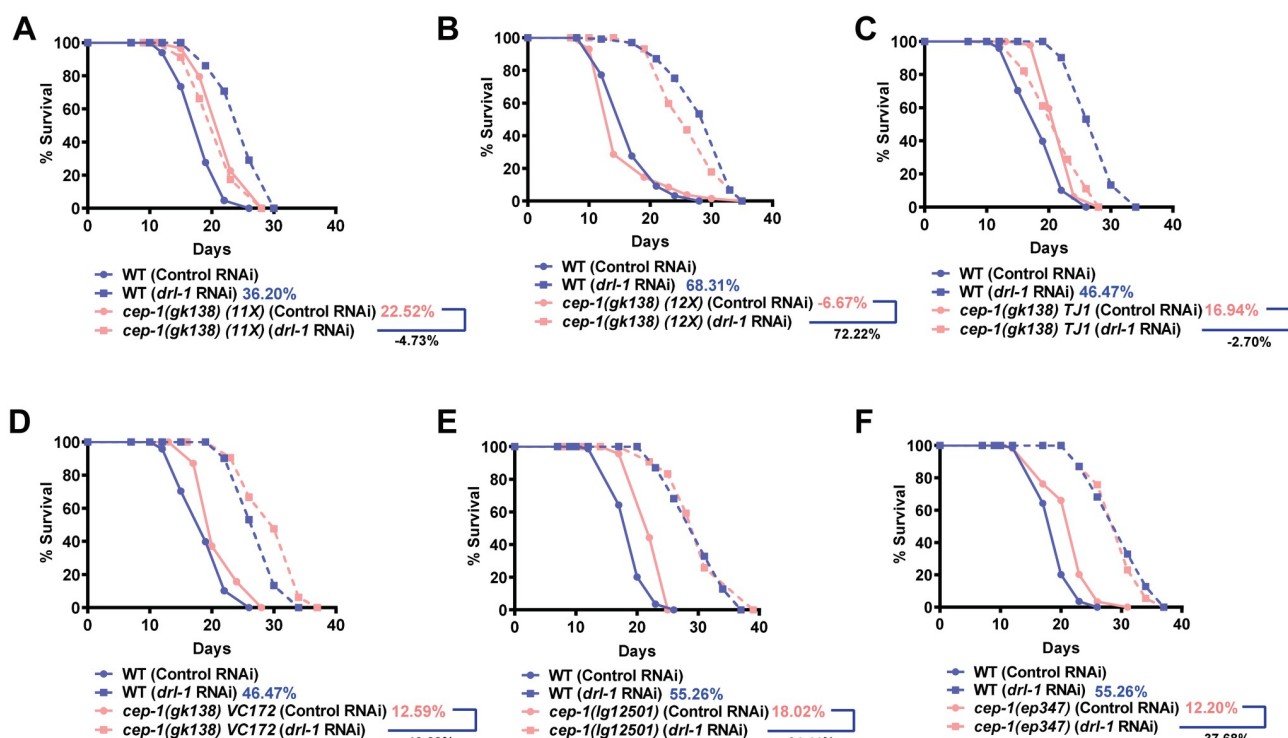

**Fig 6. Background mutation(s) in *cep-1(gk138)* may be responsible for suppressing life span of genetic paradigms of DR. (A)** Life span analysis of TJ1 *cep-1(gk138)* 11X backcrossed strain, **(B)** 12X backcrossed strain, **(C)** TJ1 strain reordered from CGC, **(D)** VC172 *cep-1(gk138)* 0X backcrossed strain, **(E)** *cep-1(lg12501)* and **(F)** *cep-1(ep347)* along with WT, grown on control or *drl-1* RNAi. Life span was performed at 20°C and details are provided in S1 Table. Mantel-Cox log rank test using OASIS software available at http://sbi.postech.ac.kr/oasis [22] was used for statistical analysis. Life spans in Fig 6C-6D and 6E-6F have identical controls as they were part of the same experiment (**also see** S4 Fig).

diverse stresses like DNA damage, oxidative stress, and nutrient deprivation. P53 regulates aging in a wide range of species, ranging from flies, worms, mice and humans [26]. As a guardian of the genome, it protects against malignancy, but this tumour suppressor activity comes as a trade-off to accelerated aging. Transgenic mice overexpressing p44, the short isoform of p53 shows reduced incidences of cancer, but with premature aging phenotypes [17]. On the other hand, the "super p53" mice model with constitutively activate p53 has lesser risk of cancer but shows no longevity benefits [27]. In *Drosophila melanogaster* (Dm), reduction of Dmp53 activity by expressing the dominant negative (DN) form of p53 in insulin producing cells (IPCs), a subset of neurons (equivalent to mammalian pancreatic beta- cells) leads to lower levels of dILP2 mRNA as well as reduced IIS pathway in fat body and thus, extends life span [19]. Life span extension by DN-Dmp53 also requires 4E-BP, a downstream component of mTOR pathway [18]. Interestingly, p53 exhibits the phenomenon of antagonistic pleiotropy in a development stage-specific and sex-specific manner. It favours survival at younger stages by acting as a tumour suppressor while it promotes cellular senescence at later stages of life, limiting survival [28]. A delicate balance in the activation status of p53 and a threshold of stress may determine the transition from favourable to detrimental effects on longevity. In addition, flies expressing DN-p53 do not show further life span extension either by calorie restriction or by over-expressing dSir2, suggesting the involvement of p53 in the calorie restriction pathway [29].

Knocking down the *C. elegans* ortholog of p53, *cep-1* leads to increased life span in a DAF-16/FOXO-dependent manner [21]. The extended life span of *cep-1(gk138)* mutant is dependent on the autophagy gene, *bec-1* [30]. In addition, *cep-1* inactivation in the mitochondrial ETC mutants shortens the life span of the long-lived *isp-1* mutant exhibiting severe mitochondrial stress while extending life span in response to mild stress, as in the short-lived *mev-1* mutant [20].

In this study, we investigated the role of the *C. elegans* ortholog of p53, *cep-1*, in DR-mediated longevity assurance. We employed the commonly-used allele, TJ1 *cep-1(gk138)* that is 10X backcrossed and available from CGC. Although we found that the strain prevented genetic modes of DR from increasing life span, the effect may be the result of background mutation(s). The TJ1 strain has been widely used to study the role of *cep-1* in aging in *C. elegans*. The TJ1 strain has a significantly enhanced life span [21] and the increased life span was dependent on functional autophagy [31]. Ventura et al., also used TJ1 to show that *cep-1* mediates the life span effects of the mitochondrial mutants, depending on the mitochondrial bioenergetic stress [32]. In the TJ1 strain, mild genetic mitochondrial perturbations that are known to increase life span failed to do so. Additionally, the life span shortening effects of severe mitochondrial disruption also required *cep-1* [32]. The TJ1 strain was also used in a later study which further characterized the opposing effects of *cep-1* on life span of mitochondrial mutants [20]. However, in that study the TJ1 strain was backcrossed two times with wild-type.

P53 is a complex biological molecule that has both pro- as well as anti-aging properties that depend on the physiological context [33]. It is the dominant tumour suppressor that facilitates DNA repair by halting cell cycle. It also directly impacts the activity of DNA repair systems [34]. Understandably, loss of p53 would lead to accumulation of mutations in the DNA. The TJ1 *cep-1(gk138)* allele may have accumulated unlinked mutations due to the absence of the p53 ortholog. As a result, the strain may show additional phenotypes that are not linked to *cep-1*. Although *Drosophila* studies indicate that DmP53 may increase life span in a manner similar to caloric restriction [18, 29], our study in *C. elegans* suggests that it may be due to background mutation(s). Considering our results, the role of p53 in DR may not have evolved in *C. elegans* or other nematodes. In future, one has to carefully design experiments to decouple the role of background mutation(s) in the highly mutable strain lacking p53, to decipher the real biological function of this important transcription factor on longevity. It will be prudent to

experiment with additionally backcrossed TJ1 as well as validating the outcomes parallelly in other alleles like *cep-1(lg12501)* and *cep-1(ep347)*.

## Experimental procedures

### Strains

All the strains were maintained at 20˚C, unless otherwise stated, on a lawn of *E. coli* (OP50) bacteria seeded on standard Nematode Growth Media (NGM) plates [35]. The *E. coli* bacteria were grown overnight in Luria Bertani (LB) media at 37 ˚C, and 200 or 1000 μl of the culture was seeded on 60 or 90 mm NGM agar plates, respectively. The plates were set at room temperature for 2–3 days to allow the bacteria to grow.

**Strains used in the study are.** N2 Bristol as wild-type, TJ1 *cep-1(gk138) I, eat- 2(ad1116) II, eat-2(ad465) II, daf-2(e1370) II, adIs2122 [lgg-1p::GFP::lgg-1 + rol-6(su1006)],* CY573-*bvIs5 [cyp-35B1p::GFP + gcy-7p::GFP],* XY1054 *cep-1(lg12501) I,* VC172 *cep-1(gk138) I,* CE1255 *cep-1(ep347).*

**Double mutants generated in the study are.** *eat-2(ad1116);cep-1(gk138),eat-2(ad465); cep-1(gk138), daf-2(e1370);cep-1(gk138), cep-1(gk138);adIs2122, eat-(ad1116); cep-1(gk138); adIs2122, eat-(ad1116);adIs2122, daf-2(e1370); cep-1(gk138);adIs2122, daf-2(e1370);adIs2122, cep-1(gk138);bvIs5 [named cep-1(gk138);CY573].*

### Preparation of RNAi plates

Nematode Growth Media (NGM) was supplemented with 100 μg/ml ampicillin and 2 mM IPTG to prepare RNAi plates. *E. coli* HT115 was transformed with the L4440 plasmid or *drl-1* cloned in L4440 plasmid. A single colony of bacteria was grown in Luria Bertani (LB) broth containing 100 μg/ml Ampicillin and 12.5 μg/ml Tetracycline, overnight at 37˚C in a shaker incubator. Next day, this culture was used as primary inoculum for sub-culturing in a fresh batch of LB media containing 100 μg/ml ampicillin at a ratio of 1:100 and grown until $OD_{600}$ reached 0.6 at 37˚C. The bacterial cells were then harvested at 5000 rpm, 4˚C and resuspended in 1X M9 buffer (at a dilution of 1:10) containing 100 μg/ml Ampicillin and 1 mM IPTG. This bacterial suspension was then seeded onto the RNAi plates and dried at room temperature for 2 days before use.

### Life span assay

Gravid adult hermaphrodite worms were bleached, washed and the eggs collected by centrifugation. These eggs were allowed to hatch on plates containing the OP50 feed or RNAi bacteria. When they reached young adult stage, they were transferred to NGM OP50 or RNAi plates, overlaid with FUDR (final concentration of 0.1 mg per ml). Worms were scored as dead or alive by prodding them with a platinum wire every 2–3 days. Unhealthy worms or worms that crawled to the sides of the plates were censored from the population. Life span graph was plotted as percentage survival on Y axis and the number of days on the X axis. Life spans are expressed as average life span ± SEM for all the life span experiments. Life span data summary is reported in S1 Table. Some of the life span assays were performed together and as a result, have same controls. For example, Figs 1B and 2B, 6C, 6D, 6E and 6F.

### BDR life span assay

BDR was performed as published earlier [1, 2]. Briefly, *E. coli* HT115 with L4440 plasmid (referred to as control RNAi) was streaked on a LB agar plate, supplemented with Ampicillin (100 μg per ml) and Tetracycline (12.5 μg per ml) and incubated at 37˚C for 14–16 hours. A

single colony was chosen from this plate and inoculated into 200 ml LB containing Ampicillin (100 μg per ml) in a 2-litre flask and grown at 37˚C for 12 hours in an incubator shaker. The bacterial cells were pelleted by centrifugation at 5000 rpm, 4˚C for 10 minutes, which was then resuspended in S-basal-cholesterol-antibiotics solution (cholesterol 5 μg per ml, Carbenicillin 50 μg per ml, Tetracycline 1 μg per ml, Kanamycin 10 μg per ml) supplemented with 2 mM IPTG. This was then diluted to the required optical density ($OD_{600}$) using S-basal-cholesterol antibiotics solution, forming different dilutions of bacteria which were kept at 4˚C for a maximum of 2 weeks.

Gravid adult worms maintained on OP50 bacterial feed were bleached and eggs were kept on a 60 mm control RNAi-seeded NGM RNAi plate. On reaching young adult stage, FUDR (100 μg per ml) was added to each plate to prevent progeny development. After about 24 hours, approximately 10–15 worms were transferred to each well of a 12-well plate containing 1 ml of S-basal-cholesterol-antibiotics solution with FUDR (100 μg per ml). The plate was kept on a shaker for 1h to remove adhering bacteria. Meanwhile, the diluted bacterial suspensions were aliquot to separate 12-well cell culture plates, 1 ml solution per well along with FUDR at 100 μg per ml. After the worms were washed off the bacteria, 10–12 worms were moved from the S-Basal to the diluted bacterial suspension with the help of a glass pipette connected to a P200 pipette. Every 3–4 days, worms were moved to fresh bacterial solutions. Before this transfer, they were scored for movement by prodding using a platinum wire. FUDR supplementation in diluted bacterial suspension was only required for the first 8 days. Worms that did not respond to gentle prodding with a worm pick were scored as dead and removed. During experiments, the temperature was maintained at 20˚C with continuous rotation of plates at 100rpm in an incubator shaker (Innova 42 incubator shaker, New Brunswick Scientific, New Jersey, USA). Bacterial dilutions had $OD_{600}$ ranging from 3.0, 1.0, 0.5, 0.25, 0.125 and 0.015625. Life span summary is reported in S1 Table. Statistical analysis was performed using Two-way Annova.

## Life span assay with 2-deoxyglucose (2-DOG)

Plates for performing 2-DOG life span were prepared from the same batch of NGM agar as the control plates except that the 2-DOG (Sigma-Aldrich, D8375) was added in the media to a final concentration of 5 mM from a sterile 0.5 M stock solution made in water. Synchronized egg population (100–150) obtained by sodium hypochlorite treatment of gravid worms grown on *E. coli* OP50, was exposed to control plates (without 2- DOG). At YA stage, approximately 50 worms were transferred to the plates containing 2-DOG and plates without 2-DOG, overlaid with FUDR, in three technical replicates. Worms were scored every alternate day as mentioned above.

## Dauer assay

Gravid adult worms were bleached and eggs were kept on OP50-seeded plates and upshifted to 22.5˚C in an incubator. After 72h, worms were counted as adults or dauers. Percentage dauers for each strain was calculated and plotted. Two biological repeats, each with at least two technical repeats, were used for averaging.

## RNA isolation, cDNA synthesis, and quantitative real time PCR

Eggs were synchronized by overnight starvation. These L1s were then allowed to grow till YA stage. YA worms were collected in M9 buffer after washing thrice to get rid of bacteria and then frozen in Trizol (Invitrogen, USA). The frozen worms were passed through two freeze-thaw cycles and lysed by vigorous vortexing. RNA was extracted by phenol:chloroform:isoamyl

alcohol followed by ethanol precipitation. RNA was dissolved in DEPC-treated MQ water and denatured at 65˚C for 10 min. The concentration of the RNA was then determined using NanoDrop 2000 (Thermo Scientific, USA). The integrity of the RNA was checked by electrophoresis on a denaturing formaldehyde gel.

First strand cDNA was synthesized using 2.5 μg RNA, employing Superscript III Reverse Transcriptase (Invitrogen, USA). Quantitative real time PCR (qRT-PCR) reaction was set up using DyNAmo Flash SYBR Green master mix (Thermo Scientific, USA) in a Realplex PCR system (Eppendorf, USA), according to manufacturer's specifications. The gene expression was represented as relative fold change determined after normalizing the ΔCt values to *actin*, the housekeeping gene. Two-way Annova with Sidak's multiple comparisons test and Unpaired two-tailed *t*-test was applied as statistical analysis by using GraphPad Prism (Graph-Pad Software, La Jolla California).

## Autophagosome quantification

Gravid adult worms expressing *lgg-1*::*gfp* were bleached and eggs were hatched on OP50 or RNAi bacteria. When they reach the L3 larval stage, worms were anesthetised with 20 mM sodium azide on 2% agarose slides and imaged at 630X using an AxioImager M2 microscope fitted with Axiocam MRc camera (Carl Zeiss, Germany). The autophagosomes were manually scored as GFP puncta in the hypodermal seam cells, seen only at L3stage. At least 15 worms were examined for approximately 3–10 seam cells in each one of them. The total number of puncta per seam cells for each worm was calculated and averaged out. Assay was repeated three times.

## Western blotting

Ten micrograms of protein for each sample was resolved on a 15% SDS-PAGE, transferred onto a PVDF membrane (Millipore, Billerica, MA) and blocked with 5% w/v skimmed milk protein prepared in 0.1% TBST (room temperature, 1 hr). Subsequently, the membrane was incubated overnight with anti-GFP antibody (Novus, USA—Cat. No. NB100-2220; 1:1000 dilution in 0.1% TBST) at 4˚C on a rocker-shaker. Following three washes of 5 minutes each in 0.1% TBST, anti-mouse secondary antibody (Abcam, UK—Cat. No. ab6728; 1:5000 dilution in 0.1% TBST) was added to the membrane and incubated for 1 hour at room temperature, with rocking. The membrane was then washed three times with 0.1% TBST, each for 5 minutes, at RT on a rocker-shaker. The blot was developed using an ECL reagent (Millipore, USA), according to manufacturer's instructions. Uncropped blots are provided in the Supplementary Information.

## Quantification of fat content by Oil Red O staining

Fat content of worms was determined according to previously published protocols [2, 36]. Prepared in advance (Oil Red O working solution): Oil Red O stain was prepared as 5mg/ml stock in isopropanol and equilibrated on a rocker shaker for a week. The working stock of Oil Red O was prepared by diluting equilibrated stock to 60% with water. Stock was mixed thoroughly and filtered using a 0.22μm filter to remove any particles.

Eggs extracted from gravid adults by sodium hypochlorite treatment were grown till L4 or YA stage. The worms were washed in 1X PBS to remove any attached bacteria and resuspended in 120μl 1X PBS. To this an equal volume of 2X MRWB buffer (160mM KCl, 40mM NaCl, 14 mMNa2EGTA, PIPES pH 7.4, 1mM Spermidine, 0.4mM Spermine, 2% Paraformaldehyde, 0.2% β-mercaptoethanol) was added and incubated for 45 minutes on a rocker shaker. The worms were stored at -80˚C after slow freezing using liquid nitrogen. For staining, the

frozen worms were thawed on ice, then pelleted and washed thrice with 1XPBS. An 120μl aliquot of the working solution of Oil Red O was added to the fixed worms and incubated for an hour on a shaker at room temperature. Following staining, worms were washed thrice with 1X PBS and mounted on 2% agarose slides for visualization using an AxioImager M2 microscope fitted with Axiocam MRc camera. ImageJ software was used to quantify fat stores in the intestine of worms.

## Statistical analysis

All *p*-values were calculated (by unpaired two-tailed *t*-test or two-way Annova) and graphs were plotted using Graph pad Prism. For life span analysis, Mantel-Cox log rank test using OASIS software available at http://sbi.postech.ac.kr/oasis [22] was used.

## Supporting information

**S1 Fig. The percentage of dauer formed in *daf-2(e1370)* is enhanced in the genetic double *daf-2(e1370);cep-1(gk138)*.** Dauer assay was performed at 22.5 ˚C. Data are presented as mean values ± SEM. N = 2 independent experiments. Unpaired two-tailed *t*-test was used for statistical analysis. **$P \leq 0.01$.
(PDF)

**S2 Fig.** Representative images showing autophagosomes as GFP puncta in the hypodermal seam cells of L3-staged **(A)** *lgg::gfp* and *cep-1(gk138);lgg::gfp* worms on control or *drl-1* RNAi, **(B)** *eat-2(ad116);lgg-1::gfp* and *eat-2(ad116);cep-1(gk138);lgg-1:gfp*, **(C)** *daf-2(e1370);lgg-1::gfp* and *daf-2(e1370);cep-1(gk138);lgg-1::gfp*. Images were captured at 630X magnification. Inset images represent zoomed-in areas showing one seam cell. Arrows point to autophagosome puncta. Scale bar is 10 μm. **(D)** The increased autophagosome formation in *daf-2(e1370);lgg-1:gfp* is unaffected in *daf-2(e1370);cep-1(gk138);lgg-1::gfp*. Quantification of GFP puncta for one of two independent experiments is shown. Data are presented as mean values ± SEM. N = 2 independent experiments. No. of animals analysed n≥17. Unpaired two-tailed t-test with Welch's correction was used for statistical analysis. ****$P \leq 0.0001$, ns = non-significant. Source data is provided as source data file. **(E)** Western blot using anti-GFP antibody to detect the unmodified LGG-1 or PE-LGG-1 in *daf-2(e1370);lgg-1::gfp* and *daf-2(e1370);cep-1(gk138);lgg-1::gfp*. β-ACTIN was used as a loading control. One representative blot out of four independent experiments is shown. Experiments were performed at 20˚C. Uncropped blots are provided in Supplementary Information file.
(PDF)

**S3 Fig. Oil Red O staining of *eat-2(ad465)* or *eat-2(ad465);cep-1(gk138)*.** Lowering of fat storage in *eat-2(ad465)* is attenuated in *eat-2(ad465);cep-1(gk138)*. Representative images (upper panel) and quantification (lower panel) for one of four independent experiments is shown. Images were captured at 400X magnification. Scale bar is 20 μm. Data are presented as mean values ± SEM. N = 2 independent experiments. No. of animals analysed n≥20. Unpaired two-tailed *t*-test with Welch's correction was used for statistical analysis. ****$P \leq 0.0001$. Experiments were performed at 20 ˚C. Source data is provided as source data file.
(PDF)

**S4 Fig. (A)** Life span analysis of *cep-1(gk138)* (12X-AB) generated earlier [20] as well as WT, grown on control or *drl-1* RNAi. Life span was performed at 20˚C and details are provided in S1 Table. Mantel-Cox log rank test using OASIS software available at http://sbi.postech.ac.kr/oasis [22] was used for statistical analysis. **(B)** The percentage of dauer formed in *daf-2(e1370)* is enhanced in the genetic double *daf-2(e1370);cep-1(gk138) 12X*. Dauer assay was performed

at 22.5 ˚C. Data are presented as mean values ± SEM. N = 3 independent experiments. Unpaired two-tailed *t*-test was used for statistical analysis. $^{**}P \leq 0.01$.
(PDF)

**S1 Table. Summary of life span analysis, related to Figs 1, 2 and 6 and S4 Fig.**
(XLSX)

**S2 Table. List of primers used in the study, related to Figs 3 and 5.**
(DOCX)

**S1 File. Supplementary information.** This file contains source data for experimental replicates used in the study and shown in Figs 3A, 3B, 4A and 4B, as well as S2D and S3 Figs. It also contains uncropped blots used in Fig 3D and S2E Fig.
(XLSX)

**S1 Raw images.**
(PDF)

## Acknowledgments

We would like to thank the members of the Molecular Aging lab for their scientific inputs and Dr. Sylvia Lee for reagents. Some strains were provided by the Caenorhabditis Genetics Center, University of Minnesota, USA.

## Author Contributions

**Conceptualization:** Arnab Mukhopadhyay.

**Formal analysis:** Anita Goyala.

**Funding acquisition:** Aiswarya Baruah, Arnab Mukhopadhyay.

**Investigation:** Anita Goyala.

**Methodology:** Anita Goyala.

**Project administration:** Aiswarya Baruah, Arnab Mukhopadhyay.

**Supervision:** Arnab Mukhopadhyay.

**Validation:** Anita Goyala.

**Writing – original draft:** Arnab Mukhopadhyay.

**Writing – review & editing:** Anita Goyala, Aiswarya Baruah, Arnab Mukhopadhyay.

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
