## [Decision Letter · Decision Letter 0]

20 Aug 2020

PONE-D-20-21394

The genetic paradigms of dietary restriction fail to extend life span in cep-1(gk138) mutant of C. elegans p53 due to possible background mutations

PLOS ONE

Dear Dr. Mukhopadhyay,

Thank you for submitting your manuscript to PLOS ONE. After careful consideration, we feel that it has merit but requires revision before it is accepted for publication. Therefore, we invite you to submit a revised version of the manuscript that addresses the points raised during the review process.

Specifically, the concerns raised by Reviewer 1 may be addressed. 

I, however, do not agree with some of the concerns raised by Reviewer 2; particularly, I believe that mapping of the background mutation, as asked by the reviewer, may not be within the scope of study. The finding that a background mutation maybe responsible for the observed effects deserves publication and is accordingly recommended after other concerns are addressed. The quantification of lipids, as pointed out by Reviewer 2, maybe looked into. It is also suggested to enhance the Discussion section by providing information on association of cep-1 in ageing and longevity. 

We look forward to receiving your revised manuscript.

Kind regards,

Aamir Nazir, Ph.D.

Academic Editor

PLOS ONE

Journal Requirements:

2.Thank you for stating the following in the Acknowledgments Section of your manuscript:

[The research was supported in part by the National Bioscience Award for Career

Development (BT/HRD/NBA/38/04/2016) and SERB-STAR award (STR/2019/000064) to AM, DBT’s

Twinning Programme for the NE (BT/PR16823/NER/95/304/2015) to AM and AB, and core funding

from the National Institute of Immunology to AM. Some strains were provided by the Caenorhabditis

Genetics Center, which is funded by National Institute of Health Office of Research Infrastructure

Programs (P40 OD010440).]

 [The funders had no role in study design, data collection and analysis, decision to publish, or preparation of the manuscript.]

Additional Editor Comments (if provided):

The concerns raised by Reviewer 1 may be addressed.

I, however, do not agree with some of the concerns raised by Reviewer 2; particularly, I believe that mapping of the background mutation, as asked by the reviewer, may not be within the scope of this study. The finding that a background mutation maybe responsible for the observed effects deserves publication and is accordingly recommended after other concerns are addressed. The quantification of lipids, as pointed out by Reviewer 2, maybe looked into. It is also suggested to enhance the Discussion section by providing further information on association of cep-1 in ageing and longevity, which will make it further enriching for the reader and help in correlating the findings.

The manuscript will be recommended for publication after revision.

Reviewers' comments:

Reviewer's Responses to Questions

**Comments to the Author**

1. Is the manuscript technically sound, and do the data support the conclusions?

Reviewer #1: Yes

Reviewer #2: Partly

2. Has the statistical analysis been performed appropriately and rigorously? 

Reviewer #1: Yes

Reviewer #2: Yes

3. Have the authors made all data underlying the findings in their manuscript fully available?

Reviewer #1: Yes

Reviewer #2: Yes

4. Is the manuscript presented in an intelligible fashion and written in standard English?

Reviewer #1: Yes

Reviewer #2: Yes

5. Review Comments to the Author

Reviewer #1: Summary: In the current manuscript, Goyala et al. investigate the role of a C. elegans ortholog of p53, cep-1, in DR-mediated longevity assurance. The authors show TJ1 cep-1(gk138) strain prevents genetic modes of DR from increasing life span, the effect may be the result of background mutation(s). This study is well designed, and the findings are clear. However, there are still some concerns that need to be clarified.

Minor

1. There are several typos and grammatical mistakes throughout the manuscript. The authors should check the manuscript carefully. A few examples are listed below:

• Line 25, Please remove the word “etc” and mention the names. Also correct/rewrite the sentence “Including an increase in expression of cytoprotective genes, better proteostasis, etc.”

• Please rewrite the sentence “We find that in cep-1(gk138), two aspects of DR, increased autophagy and the up-regulation of expression of cytoprotective xenobiotic detoxification program (cXDP) genes are dampened”

• Line 44, “Research over the past decades, many using Caenorhabditis elegans”, Please remove many.

• Line 60, “For e.g.,” Remove e.g. and replace it with example.

2. Please add statistical information in Figure legend 1 and 6.

3. Figure 3, quantification of GFP puncta averaged over 2 biological repeats. The authors should perform at least 3 biological repeats.

4. Figure S2 A-C, and 5C, please add a scale bar and mention the protein band size for figure S2E.

5. Authors have used “Fig” and “Figure” both. Please make it uniform throughout the manuscript. For example, Fig 4 and figure 5.

6. Figure 4, quantification of Oil Red O staining averaged over two biological replicates, please add at least one more replicate.

7. Please separate the main figures/legends and supplementary. For example, first add main figures 1-6, followed by supplementary figures 1-4.

8. Please add details for all Experimental procedures such as for C.elegans culture.

9. The molecular weight marker in all blots is missing. The authors should provide full-length western blots in supplementary and mention the protein band size.

Reviewer #2: The manuscript by Goyala et al examines the role of CEP-1, ortholog of mammalian p53, in the regulation of dietary restriction (DR) mediated increase in life span of C. elegans. The authors show that CEP-1 differentially regulates longevity induced by DR through genetic means versus diet or bacterial dilution. This has been shown before and mechanisms behind this have been elucidated in some detail already. Some of these have been referenced in the introduction part of the manuscript. Many papers have been published linking CEP-1 to autophagy and to mutations in electron transport chain. Authors use well established assays to study autophagy, neutral lipid levels and longevity in context of cep-1 mutation in this study. The authors contend that a background mutation in cep-1 mutant is responsible for phenotypes linked to autophagy and DR mediated longevity. I agree with authors that a background mutation might well be responsible for the difference in phenotype observed in 10X backcrossed versus 12X backcrossed cep-1 mutant. Unfortunately, the authors make no attempt to identify the background mutation say by genomics approaches followed by validation using RNAi nor do they attempt to segregate the mutation from cep-1 mutation by crosses and QTL analyses. Without identifying the mutation (one or more) and their direct effect on mechanism of longevity, this study does not add any value to the community interested in longevity and DR specifically or C. elegans biology in general. Therefore, this study does not merit publication.

P53 is considered the guardian of the genome and it is not difficult to envisage that mutations may arise at a higher frequency in cep-1 mutant than in wild type strain. Going forward, this gives the authors an excellent opportunity to study mutation(s) due to the lack of CEP-1 activity.

Other Comments

1. Life span assays: None of the life span experiments have been performed more than 2 times. Authors must perform these experiments at least three and preferably more number of times.

2. Figure 4: Discrepancy between ORO stained worms and their quantification. drl-1 RNAi and eat-2 mutation appears to cause almost complete loss of ORO stained lipids but the quantification indicates that 50-60% of lipid are present. The authors must check their method for quantification and quantify lipids biochemically.

6. PLOS authors have the option to publish the peer review history of their article (what does this mean?). If published, this will include your full peer review and any attached files.

Reviewer #1: No

Reviewer #2: No

---

## [Author Response · Author response to Decision Letter 0]

12 Oct 2020

Reviewer #1: 

Minor

1. There are several typos and grammatical mistakes throughout the manuscript. The authors should check the manuscript carefully. A few examples are listed below:

We have carefully gone through the manuscript to detect typos and grammatical errors. 

• Line 25, Please remove the word “etc” and mention the names. Also correct/rewrite the sentence “Including an increase in expression of cytoprotective genes, better proteostasis, etc.”

We have removed the word ‘etc.’ and rewritten the sentence.

• Please rewrite the sentence “We find that in cep-1(gk138), two aspects of DR, increased autophagy and the up-regulation of expression of cytoprotective xenobiotic detoxification program (cXDP) genes are dampened”

We have rewritten the sentence.

• Line 44, “Research over the past decades, many using Caenorhabditis elegans”, Please remove many.

We have removed the word, “many”.

• Line 60, “For e.g.,” Remove e.g. and replace it with example. 

We have replaced ‘e.g.’ with ‘example’.

2. Please add statistical information in Figure legend 1 and 6.

We have added detailed statistical information in the legends, including corrections for multiple testing.

3. Figure 3, quantification of GFP puncta averaged over 2 biological repeats. The authors should perform at least 3 biological repeats.

The data is now shown for one of three experiments, using scatter plot. The source data for the other replicates is provided in Supplementary information as S1 File.

4. Figure S2 A-C, and 5C, please add a scale bar and mention the protein band size for figure S2E.

We have added scale bars in the representative images and protein size marker on the blots. Such bars could not be added to Figure 5c as the image was reconstructed by stitching multiple frames.

5. Authors have used “Fig” and “Figure” both. Please make it uniform throughout the manuscript. For example, Fig 4 and figure 5.

We now uniformly use “Fig”, as suggested by the journal office.

6. Figure 4, quantification of Oil Red O staining averaged over two biological replicates, please add at least one more replicate.

The data is now shown for one of three experiments, using scatter plot. The source data for the other replicates is provided in the Supplementary Information S1 file.

7. Please separate the main figures/legends and supplementary. For example, first add main figures 1-6, followed by supplementary figures 1-4.

We have separated the legends for the main figures from the supplementary ones and have followed the Journal guidelines.

8. Please add details for all Experimental procedures such as for C. elegans culture.

We further detailed the C. elegans strain maintenance. Experimental procedures are mentioned in sufficient details to replicate the experiments. 

9. The molecular weight marker in all blots is missing. The authors should provide full-length western blots in supplementary and mention the protein band size.

We have provided the full-length blots with molecular weight marker in the Supplementary Information S1 file as source data. The figures in the main text has been modified to show molecular weight markers.

Reviewer #2: 

1. Life span assays: None of the life span experiments have been performed more than 2 times. Authors must perform these experiments at least three and preferably more number of times.

We have added the third biological replicate for most of the life spans in S1 Table.

2. Figure 4: Discrepancy between ORO stained worms and their quantification. drl-1 RNAi and eat-2 mutation appears to cause almost complete loss of ORO stained lipids but the quantification indicates that 50-60% of lipid are present. The authors must check their method for quantification and quantify lipids biochemically.

We have provided representative, contrasting images in the figures. To get a clear idea, we are now showing one of three experiments in the main figure using scatter plot to represent the distribution of the readings. The source data for the other replicates is provided in the Supplementary Information file. ORO staining of fixed worm samples is an established procedure to measure fat stores and is extensively used in the field to show differential regulation of fat metabolism in diverse strains.

---

## [Editor Report · Decision Letter 1]

16 Oct 2020

The genetic paradigms of dietary restriction fail to extend life span in cep-1(gk138) mutant of C. elegans p53 due to possible background mutations

PONE-D-20-21394R1

Dear Dr. Mukhopadhyay,

We’re pleased to inform you that your manuscript has been judged scientifically suitable for publication and will be formally accepted for publication once it meets all outstanding technical requirements.

Kind regards,

Aamir Nazir, Ph.D.

Academic Editor

PLOS ONE
---

## [Editor Report · Acceptance letter]

26 Oct 2020

PONE-D-20-21394R1 

The genetic paradigms of dietary restriction fail to extend life span in *cep-1(gk138)* mutant of *C. elegans* p53 due to possible background mutations 

Dear Dr. Mukhopadhyay:

I'm pleased to inform you that your manuscript has been deemed suitable for publication in PLOS ONE. Congratulations! Your manuscript is now with our production department. 

Kind regards, 

on behalf of

Dr. Aamir Nazir 

Academic Editor

PLOS ONE